# Climate Change Modulates Halophyte Secondary Metabolites to Reshape Rhizosphere Halobacteria for Biosaline Agriculture

Asadullah [1] and Asghari Bano [2],*

1    Department of Biology, The Peace Group of Schools and Colleges, Charsadda 24420, Pakistan
2    Department of Biosciences, University of Wah, Wah Cantt 47010, Pakistan
*    Correspondence: asghari.bano@uow.edu.pk

**Abstract:** To feed the ever-increasing population under changing climate scenarios, it is imperative to investigate the role of halophytes, which are equipped with special adaptation mechanisms to cope under extreme conditions of salinity. In the current review, we aimed to report newly identified bioactive secondary metabolites that might play a role in establishing rhizosphere microbe associations, elucidate the negative impacts of salt stress, and direct the growth and yield of halophytes. A systematic approach was developed that deciphers those metabolites involved in regulating the physiological, biochemical, and molecular responses of halophytes to salt stress. The mechanism of salinity tolerance, recruitment of beneficial microbes, and signaling role of secondary metabolites were also discussed. The role of halotolerant rhizobacteria' secondary metabolites in the physiology and growth parameters of halophytes was also discussed.

**Keywords:** climate change; rhizosphere microbes; secondary metabolites; halophytes





## 1. Introduction

### 1.1. Impact of Climate Change on Biosaline Agriculture and the Role of Halophytes

Current climate change poses a serious threat to agricultural productivity. A rise in the atmospheric $CO_2$ level, heat waves, elevated temperature, soil drought, and salinity are all major outcomes of climate change that adversely affect the growth of glycophytes [1,2]. Halophyte crops are better alternatives for food, fodder, fiber, fuel, essential oil, and medicines, as they have developed special characteristics to cope with environmental extremes. Halophytes possess a greater degree of tolerance than non-halophyte crops. Some parasitic plants develop succulence when growing on halophytic hosts, and an interesting finding is that halophytes growing in their natural habitats do not show signs of oxidative stress [3]. *Halogeton glomeratus* is an annual herbaceous and succulent halophyte belonging to the family Chenopodiaceae. It is considered a potential source for oilseed production and also assists in phytoremediation. Exposure of *H. glomeratus* to long-term salinity and drought stress results in decreased chlorophyll and carotenoid content and inhibition of the photosynthetic rate, transpiration rate, stomatal conductance, water potential, and biomass [4].

Halophyte plants survive in soil where salinity is around 200 mM NaCl [5]. Plant ecologists classify halophytes into three main groups. The first is euhalophytes that dilute salt within their stems or leaves and have a strong capability to endure salt stress. Recretohalophytes secrete salt from their leaves and grow widely around the world, inhabiting inland saline lands and seawater. Pseudo-halophytes can not only hold up ions in roots but also minimize their transport to the shoot parts. On the basis of morphology, they are classified into two groups: excretive and succulent. Based on salt demand and tolerance to NaCl, they can be classified as obligate halophytes, facultative halophytes, and habitat-indifferent halophytes. Obligate halophytes are true halophytes, as they need salt for their growth. Members of the family Chenopodiacea belong to this category. Facultative halophytes generally grow in soil where salts are in quite low concentrations, but they can also grow under

saline conditions. Monocots belonging to families Poaceae, Cypraceae, and Juncaceae and a large number of dicotyledons belong to this group. Habitat-indifferent halophytes prefer to live in saline-free soil but can also thrive in saline soil. Examples are *Festuca rubra, Agrostis stolonifera,* and *Juncus bufonius.* On the basis of habitat, halophytes are classified into hydrophalophytes (grow in saline water or salt marshes) and xerohalophytes (grow in desert). Salinity induces the release of oleanolic acid in the root exudates of *Salicornia*, which act as chemoattractants for colonizing halophilic siderophores producing *Halomonas anticariensis* FP35$^T$ [6]. In return, colonization of this bacterium enhances the positive effects on the root length, shoot length, germination, and vigor index of *S. hispanica* [6].

*1.2. Halophytes as Crop Plants*

For agricultural sustainability under saline conditions, two possible approaches are needed: (i) improving salt tolerance of cultivated crops, or (ii) domestication of halophytes [5]. The majority of crop plants are non-halophytes. Genetically modified crops have been developed for biosaline agriculture, but this is time consuming and involves many genes possessing various pros and cons and causes various allergic reactions, disrupting natural gene flow and increasing the risk factors of human diseases [7].

The cultivation of halophytes for food, fodder, edible oil, biofuel, and medicine seems to be an alternate option, as they are fully equipped with better salt tolerance properties. Many species of halophytes have the potential to be used as gourmet vegetables and salad, i.e., *Aster tripolium, Atriplex hortensis, Beta maritime, Crambe maritima, Crithmum maritimum, Inula crithmoides, Salicornia* spp., *Salsola soda*, and *Tetragonia tetragonioides*. Some wild edible halophytes are in the genera *Bassia, Beta, Cakile, Chenopodium, Plantago, Portulaca*, and *Suaeda*. *Atriplex halimus, Salicornica fruticosa*, and *Cakile maritime* accumulate Na$^+$ in their leaves, but the content of this ion is lower compared to other culinary halophytes. These plants can be used as sources of green salt (plant-based salt contains 50 % less Na than common salt) due to their nutritional value [8,9]. The seeds and lignocellulosic biomass of several halophyte genera, i.e., *Salicornia, Suaeda, Atriplex, Distchlis*, and *Batis,* have been exploited for the production of biofuel (bioethanol). *Suaeda* is a high salt tolerant C$_4$ species. This genus ameliorates contaminated saline soil and is also used in food, fodder, medicine, and bioenergy.

A wide group of halophyte species (2500–3000) occur naturally in salt marshes, providing an opportunity to use them as crop plants to combat food security in the future. Seven species of plants, namely *Suaeda glauca, Bassia scoparia, H. glomeratus, Kalidium foliatum, Medicago falcata, Atriplex canescens*, and *Artemisia desertorum,* can be used as enrichment materials for Zn and Cu. *H. glomeratus* has medicinal value in traditional Chinese medicine. Wang et al. [10] identified secondary metabolites that included flavones, flavonols, flavandiols, glucosinolates, isoquinolines, pyridines, indoles, amino acids, lipids, carbohydrates, and ATP-binding cassette transporters. These metabolites regulate osmotic adjustment and modulate the adjustment of membrane lipid action in *H. glomeratus*. These metabolites also have applications in human cardiovascular diseases, cancers, diabetes, and heart diseases. Economic importance of some halophytes are listed below in Table 1.

**Table 1.** Economically important halophytes, their bioactive metabolites, and their role in plants and humans.

| Species | Secondary Metabolites | Economic Value | Reference |
|---|---|---|---|
| *Apocynum venetum* | Flavonoids | Anti-inflammatory activity | [11] |
| *Atriplex triangularis* | Hydroxyecdysone, flavonoids, and phenolics | Nutritional properties rich source of proteins, vitamin A, and vitamin C | [12] |
| *Avicennia marina* | Saponin, triterpenoid, or phytosterol. | As fodder, biological and pharmacological importance | [13] |
| *Beta vulgaris* | Phenolics, carotenoid, ascorbic acid, betalains | food additive, improve redness in tomato paste and jellies | [14] |

**Table 1.** *Cont.*

| Species | Secondary Metabolites | Economic Value | Reference |
|---|---|---|---|
| *Suaeda fruticosa* | | Effective against *B. subtilus*, *P. aeruginosa*, and *S. aureus* | [15] |
| *Cenchurus cilliaris* | Stigmasta-5, 22-Dien-3-ol, 6,6-Dideutero-nonen-1-ol-3 | Used in Pharmaceutical Industry | [16] |
| *Chenopodium quinoa* | Phytoecdisteroids and polyphenols | Seed protein content twice that of rice, contains lysine | [17] |
| *Lespedeza bicolor* | Ptrerocarpans, lespedezol, dihydrolespedezol | Significant virucidal activity | [18] |
| *Suaeda vermiculata*, *Salsola cyclophylla* | Camphor, benzoic acid ester, borneol, α Terpineol, hexahydrofarnesyl acetone | antimicrobial and antioxidant activity | [19] |
| *Thymus spinulosus*, *Satureja cuneifolia* | thymol, alpha pinene | as natural pesticide in organic agriculture | [20] |
| *Tamarix africana*, *Suaeda fruticosa* | Rutin, Gallic acid, Kaempferol 3-O-glucoside | bactericidal activity against *B. subtilis* and *S. auresus*, strong antifungal effect against *C. albicans* | [21] |
| *Tecticornia* sp. (Samphire) | Celossianin II, isocelosianin II, | indigenous edible halophyte in Australia | [22] |
| *Eryngium maritimum* L. | carvacrol, 2,3-dimethoxybenzoic acid, naringenin, catechin, and t-cinnamic acid | In vitro inhibition of diabetes-related enzymes, antioxidant potential | [23] |
| *Artemisia campestris* subsp. *maritima* | quinic, chlorogenic and caffeic acids, coumarin sulfates, and dicaffeoylquinic acids | Pharmaceutical, cosmetic, and/or food industries | [24] |
| *Limonium algarvense* | 52 different metabolites present in leaves extract identified through LC-ESI(-)-HRMS/MS | Antioxidant, anti-inflammatory, neuroprotective, and antidiabetic properties | [25] |
| *Limonium vulgare* | quercetin and myricetin or myricetin 3-rhamnoside | Nutraceuticals and/or pharmaceuticals | [26] |

*1.3. Salt Tolerance Mechanism of Halophytes: Role of Secondary Metabolites*

According to Meng et al. [27], halophytes have adapted to thrive under high salinity conditions by secreting salt crystals through salt glands, regulating cellular ion homeostasis and osmotic pressure, detoxifying reactive oxygen species (ROS), and bringing alterations in membrane composition. A diverse group of secondary metabolites is involved in regulating these functions. Functionally, they are classified as osmoprotectants, antioxidants, polyamines, and phytohormones. On the basis of their biosynthetic pathway, they are categorized into (i) phenolic groups (composed of single sugar and benzene rings); (ii) terpenes and steroids; and (iii) N-containing compounds. In this section, an effort has been made to explain the role of secondary metabolites in modulating the salt tolerance strategies of halophytes. The following are adaptations of selected halophytes against salt stress.

1.3.1. Modification in Morphology or Anatomy

Salt stress induces anatomical modifications in roots, stems, and leaves of halophytes. Under high salinity, *C. cilliaris* conserved water by increasing sclerification in the cortex and pith region and decreasing root thickness with a greater proportion of parenchyma cells, and rich density of vesicular hairs and trichomes on leaves might be essential for water conservation and salt excretion [28]. The pattern of adaptation is species specific and variable among different species, i.e., *Salicornia perennans* ($C_3$ sp.) showed larger variation in leaf functional traits, both at the level of cell morphology and membrane system (chloroplast envelope and thylakoid); however, *Climacoptera crassa* ($C_4$ sp.) showed an increase of the mesophyll cell surface, expansion of the interface area between mesophyll and bundle sheath cells, and an increased volume of the latter [29]. These modifications were compensated for the scarce $CO_2$ supply and increased salt concentration.

*Chloris gayana* is a $C_4$ monocot halophyte that tolerates salinity by accumulating the highest level of $Na^+$ and $Cl^-$ in xylem parenchyma and epidermal cells but maintaining the lowest level in photosynthetically active mesophyll and bundle sheath cells. Furthermore, it was lowered more than in respective vacuoles [30]. Several genotypes of *Trifolium fragiferum* were grown at different salinity levels in controlled conditions. The accumulation of mineralome was both genotypic and organ specific. Maximum $Na^+$ and $Cl^-$ were accumulated

in leaf petioles followed by leaf blades and stolons; the Zn concentration increased in all plant parts, and the $Ca^{+2}$ and $Mg^{+2}$ concentrations decreased [31]. In another study, it was reported that *T. fragiferum* showed high competitive ability based on better physiological salinity tolerance [32]. *Solanum chilense* is a halophyte sp. and wild relative of *Solanum lycopersicum* that can maintain its reproduction despite the accumulation of $Na^+$ in its floral organs [33].

### 1.3.2. SOS System Activate Salt Glands to Exclude Na from Cells

The sodium overly sensitive system (SOS) plays a crucial role in excluding $Na^+$ from roots and loading to the xylem to regulate ion homeostasis. The SOS system is composed of SOS1, SOS2, and SOS3. Under salt stress, the increase in cytosolic $Ca^{+2}$ is perceived by the SOS3 component, which interacts and activates SOS2, which finally activates the SOS1 Na/H antiporter. SOS1 then mediates $Na^+$ efflux to the apoplast through active transport driven by the H gradient across the plasma membrane established by H-ATPase.

Recretohalophytes show higher salt tolerance potential than other halophytes due to the presence of salt glands, which secrete excess salt crystals onto the leaf surface to maintain cellular ion homeostasis. The genera of recretohalophytes that show such mechanisms include *Limonium* and *Tamarix*. Ions secreted through salt glands include cations i.e., $Na^{+1}$, $K^{+1}$, $Ca^{+2}$, $Mg^{+2}$, $Mn^{+2}$, and $Fe^{+2}$, and anions ($Cl^{-1}$, $Br^{-1}$, $I^{-1}$, $SO_4^{2-}$, $PO_4^{3-}$, and $NO_3^{-1}$). *Suaeda salsa* exhibited greater expression of the *S. salsa SOS1* gene in roots, which was correlated with the increased exclusion of Na by roots [34].

### 1.3.3. Succulence Mechanism and Epidermal Bladder Cells

Halophytes exhibit selective $Na^{+1}$ transport; if the concentration is high, then it is sequestered in vacuoles. $Na^{+1}$ sequestration is possible though the NHX family Na/H antiporter in the tonoplast and V-H-ATPase, which create a pH gradient between the cytoplasm and vacuole, allowing $Na^{+1}$ to enter into the vacuole against the electrochemical gradient. The compartmentalization of $Na^{+1}$ in the vacuoles contributes not only to ion homeostasis and cell turgidity but also protects metabolic enzymes from ion toxicity. Some halophytes exhibit the succulence property as they have thick leaves and stems to accumulate excess $Na^{+1}$ and $Cl^{-1}$ in vacuoles, increasing the size of mesophyll cells, and have smaller intercellular spaces to enhance water content and cause a high turgor pressure.

Duan et al. [35] reported that *Glycine max* showed a metabolic response to salt stress by producing isoleucine, serine, and aspartic acid. Similarly, three halophytes (*Frankenia pulverulenta*, *Atriplex prostrata*, and *Plantago coronopus*) showed increased content of alkaloid derivative, polyamines (spermidine + spermine ratio) against 400 mM NaCl stress [36]. Epidermal bladder cells, also known as salt bladders, contain vacuoles that store excess $Na^+$ and $Cl^-$. They also store ROS-scavenging metabolites and organic osmoprotectants.

### 1.3.4. Osmotic Adjustment through Osmolytes

Halophytes synthesize osmolytes in response to low external water potential to adjust osmosis and maintain positive turgor pressure. Some examples of osmolytes are proline, aquaporin, and glycine betaine. *Salicornia* and *Sarcocomia* species have a long history of human consumption and are ideal models for developing halophyte crops. Osmotic adjustment in both species is related to sequestration of $Na^{+1}$, $Cl^{-1}$, $Ca^{+2}$ in the shoot that might be achieved via production of high levels of glycine betaine [37]. Yadav et al. [38] reported that amino acids, sugars, organic acids, quinidine acid, kaempferol, and melatonin contents were increased under elevated stress conditions (500 mM NaCl and 5% polyethylene glycol) for 24 h.

### 1.3.5. Regulation of ROS by Secondary Metabolites

Reactive oxygen species (ROS) play a role in the signaling pathway, but their increased accumulation in cells triggers oxidative stress. Halophytes are equipped with powerful

antioxidant systems with enzymatic and non-enzymatic components to regulate ROS in the cell. *Cakile maritima* exhibited stimulation of amino acid biosynthesis and decreased sugar content and GABA to 400 mM NaCl stress at day 20 [39]. Three wild spp. of *Suaeda* synthesize betacyanin to regulate ROS and avoid oxidative stress [40]. Salinity also induces the production of various flavonoids and phenolic compounds in halophytes against ROS.

Pungin et al. [41] reported that two halophytes, *Spergularia marina* and *Glaux maritima*, inhabiting salt marshes showed an increased content of flavonoids (hesperetin, epicatechin, apigenin derivative, luteolin derivative) and protocatechuic acid, which were also correlated with increased antioxidant activity. Linic et al. [42] reported an increased accumulation of phenolic acids, particularly hydroxycinnamic acids, and decreased accumulation of caffeic, salicylic, and 4-coumaric acid in three *Brassica* spp., upon short-term exposure to 200 mM NaCl stress. Myo-inositol, a derivative of inositol upregulated stress responsive genes in *Chenopodium quinoa,* is related to antioxidant enzyme activities and increased accumulation of free amino acids and soluble sugars under salt stress [43].

### 1.3.6. Activation of Hormonal Signaling

Abscisic acid (ABA), jasmonic acid, and ethylene are the main plant hormones produced in response to salt stress. They act as signaling molecules to activate salt tolerance strategies, i.e., halotropism is the tropic movement of Na in the roots, in which halophytes obtain optimum salt in order to maintain growth and development, while non-halophytes show negative halotropism to avoid salt stress. *Limonium bicolor* is a recretohalophyte with multicellular salt glands. This plant showed positive halotropism, as evidenced from root elongation when treated with 200 mM NaCl stress. IAA was involved in regulating this phenomenon [44]. Phytohormones also maintain the high vitality of anthers and pollens in some halophytes under salt stress. Guo et al. [45] reported a higher number of pollens and pollen activity in *Suaeda salsa* under saline conditions. GA signals partially act with JA signals via regulation of JA biosynthesis that participate in stamen development under saline conditions.

### 1.3.7. Maintaining Biogenetics

*Haloxylon salicornicum* is a xero-halophyte found in saline and arid regions of the world. Panda et al. [46] used GC-QTOF–MS and HPLC-DAD to analyze the expression of metabolites in shoot samples of *H. salicornicum* treated with 400 mM NaCl. Out of 56 identified metabolites, 47 were significantly changed in response to salt stress. These metabolites were mainly amino acids, organic acids, amines, sugar alcohols, sugars, fatty acids, alkaloids, and phytohormones. Some amino acids, e.g., alanine, phenylalanine, lysine, and tyramine, were upregulated. These metabolites upregulated the tricarboxylic (TCA) cycle by optimal production of coenzymes (NADH and FADH2) and ample quantity of ATP during stress conditions.

*Carix pumila* was treated with 200, 300, 400, and 500 mM NaCl for 60 h, and leaf samples were analyzed to identify metabolites involved in the regulation of glycolysis, the pentose phosphate pathway, and the TCA cycle [47]. Heat map of hierarchical cluster analysis revealed changes in 39 metabolites, including 16 organic acids (galactaric acid, succinic acid, benzoic acid, fumaric acid, glyceric acid, malonic acid, gluconic acid, glutaric acid, malic acid, hexadecanoic acid, propanoic acid, octadecanoic acid, phosphoric acid, hexacosanoic acid, and hydroxycarbamic acid), 9 amino acids (alanine, proline, aspartic acid, valine, serine, asparagine, glycine, threonine, isoleucine), 9 kinds of sugars (glucose, mannopyranoside, galactopyranose, glucopyranose, galactose, allose, sucrose, fructose, mannose), 3 sugar alcohols (glycerol, galactinol, myo-inositol), and 2 amines (ethanolamine, hydroxylamine) [48].

### 1.4. Halobacteria Diversity

*Halobacterium* species are obligate aerobic, rod-shaped archaea enveloped by a single lipid bilayer membrane surrounded by an S-layer made from the cell-surface glycoprotein [7]. They can use amino acids for their growth in aerobic conditions, but they can also grow in an anaerobic environment, given the correct conditions [7]. Halobacteria can be found in highly saline lakes, such as the Great Salt Lake, the Dead Sea, and Lake Magadi. Halobacteria are candidates for a life form present on Mars. These microorganisms develop a thin crust of salt that can moderate some of the ultraviolet light and make it opaque through their photosynthetic pigment bacteriorhodopsin [7].

Halobacteria are able to survive under a wide range of salinities. According to their survival potential, they are classified as halotolerant or halophilic bacteria. Halotolerant bacteria can survive in media up to 25% NaCl, whereas halophile bacteria require salt to grow. They have been isolated from different halophyte plants across the world, i.e., the rhizosphere of *Leymus chinensis*, *Puccinellia tenuiflora*, and *Suaeda glauca*, are highly enriched with the phyla of bacteria Actinobacteria, Acidobacteria, Bacteroidetes, Chloroflexi, Firmicutes, *Gemmatimonadetes*, Haloarchaea, and Proteobacteria [49–51]. The Firmicutes phylum consisted of genera *Bacillus*, *Virgibacillus*, *Salincoccus*, *Marinococcus*, *Halobacillus*, *Planococcus*, *Thalassobacillus*, and *Salimicrobium*; in phylum Actinobacteria, *Nocardiopsis* was the representative genus; in phylum Proteobacteria, *Halomonas*, *Idiomarina*, and *Psychrobacter* were representative genera [48]. Halobacteria exhibit a highly complex network of diversity than endophytic bacteria and bacteria from bulk soil. Gao et al. [52] reported Actinobacteria, Bacteroidetes, Firmicutes, and Proteobacteria in the rhizosphere of *Salicornica europaea*, *Kalidium foliatum*, and *Borszowia aralocaspica*. These bacteria exhibit a variety of mechanisms that have multiple roles, e.g., safeguard halophytes from salt stress, promote their growth, and remediate soil contamination (Figure 1).

### 1.5. Climate Change Modulates Rhizosphere Microbial Community

The main factors governing the diversity of the rhizosphere microbiome are (1) climatic factors (precipitation, temperature, salinity, drought, etc.), (2) soil physicochemical properties (pH, EC, cation exchange capacity, organic C content), and (3) root exudation pattern and composition [45]. Lashini et al. [53] isolated rhizobacteria from the rhizosphere soil of olive trees grown in semiarid and arid areas of Morocco. The strains were 85% halotolerant and 65% thermotolerant, and they were able to overcome high salinity ($\geq$4%) and temperature stress ($\geq$45 °C). They were identified as *Bacillus licheniformis*, *Arthrobacter globiformis*, and *Bacillus megaterium*. About 21% of photosynthetically fixed C is transferred into plant roots and root exudates in the form of soluble sugars, phenolic acids, amino acids, and organic acids. The elevated level of such compounds alters the community composition and structure of active bacteria [54]. The greater input of liable C via root exudates may increase microbial N demand, as competition occurred between microbes and plants for available N. Thus, N dynamics are likely to change under elevated $CO_2$. It also affects the community of N-fixing bacteria [54]. Salinity influences the structure of the rhizosphere microbiome of halophytes. Mukhtar et al. [55] conducted a study to compare the composition of the rhizosphere microbiome of halophytes *Urochloa*, *Kochia*, *Salsola*, and *Atriplex* inhabiting the moderate and high saline environment of Khewra salt rang, Pakistan, with that of non-halophyte *Triticum*. Analysis of the 16S rRNA gene showed that Actinobacteria were dominant in the saline soil. Other identified groups were Euryarchaeota, Ignavibacteriae, and Nanohaloarchaeota. Soil physicochemical properties, i.e., pH, EC, and SOC, also influence the microbe community. For example, TS, $Cl^-$, $SO_4^{2-}$, $HCO_3^{-1}$, $Na^+$, and $Mg^+$ are the main factors influencing the rhizosphere microbiome structure [56]. Soil pH is one of the key factors shaping halophytic rhizosphere soil bacterial community composition and diversity, and SWC content is a possible factor affecting bacterial community functions [57].

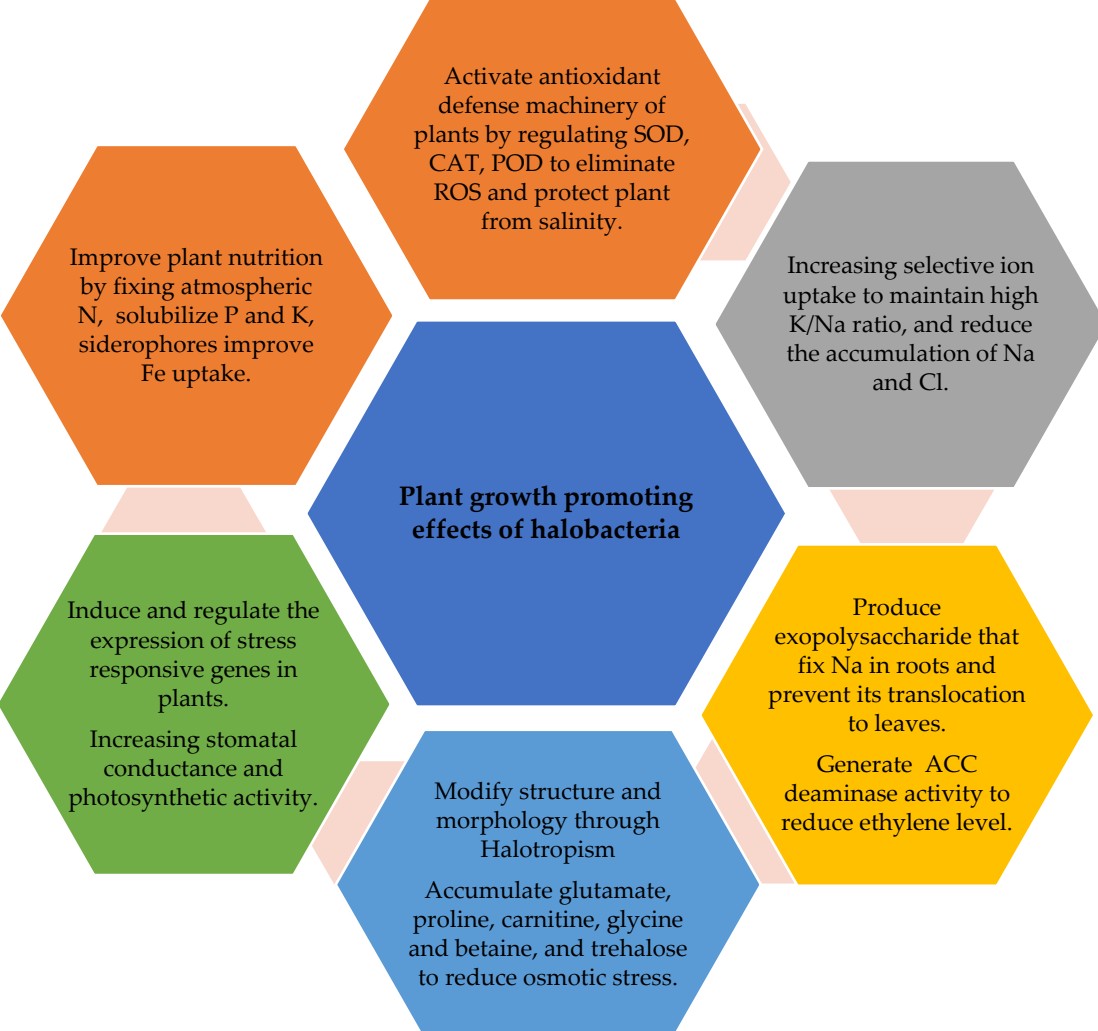

**Figure 1.** Mechanism of halobacteria in ameliorating salt stress and improving growth and yield of halophytes under saline conditions.

### 1.6. Secondary Metabolites Recruit Beneficial Rhizospheric Microbes

Root exudates are crucial in modulating the composition and functional diversity of rhizosphere microbes. Mucilage is actively released from the roots, while diffusates are passively released due to osmotic differences between the soil solution and the cell. Both types include organic compounds, which are classified into low and high molecular weight compounds. Heavy molecular weight compounds (i.e., mucilage, cellulose) are not easily used by microbes and make up the majority of C released from roots. Low molecular weight compounds are highly diverse and have a wide array of functions. These consist of organic acids, amino acids, proteins, sugar, phenolics, and other secondary metabolites (including benzoxazinoids, coumarins, flavonoids, indole compounds, and terpenes) that shape the rhizosphere microbiome [58].

Plant secondary metabolites have differential effects on soil microbiota. Two indole metabolites, benzoxazolinone and gramine, produced by different Graminae species, and quercetin, a flavonoid synthesized by many dicot species, revealed significant effects on soil bacteria, with benzoxazolinone showing a predominantly inhibitory effect preventing the accumulation of many predominantly harmful taxa, while gramine and quercetin mostly exert their function by attracting beneficial bacteria [58]. Oleanolic acid possesses chemoattractant properties and is the principal constituent of *Salicornia* root exudates

that have the potential to expand the colonization of salt-tolerant *Salicornia hispanica* by halophilic siderophore-producing bacteria *Halomonas anticariensis* FP35$^T$ [6].

Terpene is a highly diverse group of secondary metabolites that play an important role in plant microbe interaction. *Arabidopsis thaliana* root exudates contain triterpenes, namely thalianin, thalianyl medium-chain fatty acid esters (three steps), and arabidin, that are involved in belowground communication and able to directly modulate *A. thaliana*-specific root bacterial communities in a very selective manner [59]. Xiong et al. [60] reported that organic acids (2-methylbutyric acid, stearic acid, palmitic acid, palmitoleic acid, and oleic acid) found in the root exudates of *Limonium sinesis* promoted the chemotaxis of the rhizosphere PGPR strain *Bacillus flexus* KLBMP 4941. Strigolactone is a new class of phytohormone that acts as a signaling molecule in symbiosis to recruit root- and rhizosphere-associated microbiomes. Kim et al. [61] analyzed the bacterial and fungal microbial communities of 16 rice genotypes differing in exudation of root strigolactone. The results showed that structural differences in the exuded strigolactones affected different sets of microbes, i.e., the relative abundance of phosphate solubilizing microbes; *Burkholderia*, *Caballeronia*, *Paraburkholderia* and *Acidobacteria* were linked to orobanchol strigolactone, whereas 4-deoxyorobanchol was associated with genera *Dyella* and *Umbelopsis* [61].

### 1.7. Halobacteria for Biosaline Agriculture

Halobacteria possess plant growth promoting characteristics, i.e., Reang et al. [62] isolated halotolerant and halophilic bacteria belonging to *Halomonas pacifca*, *H. stenophila*, *Bacillus haynesii*, *B. licheniformis,* and *Oceanobacillus aidingensis*. These isolates were able to produce indole acetic acid, solubilized phosphate, and potash and showed N-fixing capacity. There is a growing number of publications on plant growth promotion by halophilic rhizobacteria isolated from the roots of halophyte species [63,64].

Endophytic halophiles (*Halomonas* and *Bacillus*) isolated from the roots of halophiles showed maximum salt tolerance up to 4 mM NaCl [65]. When these isolates were used to inoculate alfalfa seedlings, they stimulated plant growth in the presence of 1% NaCl, a level that significantly inhibited the growth of uninoculated plants. Exopolysaccharides (EPS) are commonly produced metabolites by halotolerant bacteria, which exhibit 40–90% extracellular matrix of bacteria under stress conditions [66]. EPS exhibits antioxidant activity and confers tolerance to bacteria against reactive oxygen species. This property can be exploited to alleviate salt stress damage in crops [67].

*Robinia pseudoacacia* seedlings exposed to VOCs of the JZ-GX1 strain showed increases in biomass, plant development, and lateral root numbers. Additionally, decreases in malondialdehyde, superoxide anion ($O_2{}^-$) and hydrogen peroxide ($H_2O_2$) contents and increases in proline contents and superoxide dismutase, peroxidase, and glutathione reductase activities were observed in *Acacia* leaves. Notably, the sodium–potassium ratio in the roots, stems, and leaves of *Acacia* exposed to VOCs of the JZ-GX1 strain were significantly lower than those in the control samples [68]. Swiss Chard inoculated with halotolerant PGPR and watered with 85 nmol L$^{-1}$ NaCl showed higher values of leaf dry weight than control plants [69]. Furthermore, PGPR inoculation reduced electrolyte leakage and Na+ uptake and improved the chlorophyll a fluorescence parameters, chlorophyll, and carotenoid concentrations, stomatal conductance, and antioxidant capacity of Swiss chard. Nijafi Zilaie et al. [70] reported that two halotolerant bacteria *Bacillus pumilus* and *Zhihengliuella halotolerans* were able to reduce the content of ascorbic acid, flavonoid, total phenol, proline, malondialdehyde, and catalase activity, and ultimately improved the antioxidant capacity of *Haloxylon aphyllum* (Table 2).

Ullah and Bano [71] reported that methanolic extracts of *Suaeda fruticosa* leaves at a concentration of 0.5 mg/mL and 0.35 mg/mL were found active against *B. subtilis*, *S. aureus*, *E. coli*, *P. aeruginosa*, *Candida tropicalis,* and *Candida albicans*. *Suaeda fruticosa* aqueous extract showed hypoglycemic (41%) and anti-hyperglycemic (53%) effects in the hypercholesterolemic and insulin-resistant sand rats. The endophytes residing in the roots of halophytes have a better adaptation to saline conditions. Hassan et al. [72] demonstrated

that application of root powder of *Cenchrus ciliaris,* a halophytic weed grown in Khewra salt range, induces salt tolerance in wheat, which was mediated by 3 PGPR, *B. cereus*, *P. moraviensis,* and *Stenotrophomonas maltophilia,* inhabiting the roots of the halophytic weed.

The *Glutamicibacter* genus is a promising candidate for phytoremediation of saline soil due to multiple potential plant growth promotion traits and tolerated a high concentration of NaCl (Table 2). *Glutamicibacter halophytocola* isolated from roots of *Limonium sinesis* significantly promoted the growth of *L. sinesis* under NaCl stress. Inoculation of *L. sinesis* with this bacteria increased the concentrations of total chlorophyll, proline, antioxidative enzymes, flavonoids, $K^+$, and $Ca^{2+}$ in the leaves; the concentrations of malondialdehyde (MDA) and $Na^+$ were reduced [73].

**Table 2.** Effect of halobacteria on the growth and physiology of some halophytes.

| Halobacteria | Host Plant | Effect on Host Plant | Reference |
|---|---|---|---|
| *Alcaligenes faecalis* SBN01 and SBN02 | Wheat | Plant biomass increased at 600 mM NaCl, accumulation of Total Chl, and Carotenoid | [74] |
| *Glutamicibacter halophytcola* | Tomato seeds | At 200 mM NaCl, Root biomass increased, K+/Na+ ratio changed | [75] |
| *Bacillus pumilus* FAB10 | Wheat | At 250 mM NaCl, internal $CO_2$ increased, reduced CAT, SOD, and glutathione reductase | [76] |
| *Aneurinibacillus aneurinilyticus*, *Paenibacillus* sp. | *Phaseolus vulgaris* | root (220%) and shoot (425%) biomass and total chlorophyll content (57%) increased | [77] |
| *Klebsiella* sp. | *Avena sativa* | Increased relative water content, proline content, electrolyte loss, MDA content in shoots, and decreased SOD and POD | [78] |
| *Bacillus tequilensis, Bacillus aryabhattai* | Rice | Increase photosynthetic rate, transpiration, stomatal conductance, grain yield | [79] |
| *B. marisflavi, Zhihengliuella flava* and *H. nanhaiensis* | *Zea mays* L. | Root and shoot length increased in 200 to 400 mM NaCl, high accumulation of proline compared with the non-inoculated plants | [80] |
| *S. chartreusis, S. tritolerans,* and *S. rochei* | *Salicornia bigelovii* | enhanced shoot and root dry biomass by 32.3–56.5% and 42.3–71.9%, respectively, 69.1% increase in seed yield | [81] |
| *Alcaligenes* sp. AF7 | Rice | enhanced the fresh and dry biomass of rice at 170 mM NaCl (EC 9 dS/m) | [82] |
| *Halomonas* sp. Exo1 | Rice | enhanced germination index upto 83%, enhanced length and weight of vegetative parts | [83] |
| *P. plecoglossicida, B. flexus,* and *B. safensis* | *Bacopa monnieri* | Increased in shoot $Na^+/K^+$ ratio, increased the growth, and increased bacoside A yield | [84] |
| *Staphylococcus* sp., | *Salicornia* sp. | At 200 MNaCl, plant growth index was increased by 13.9–47.0% | [85] |
| *Halomonas, Bacillus* | Alfalfa | Increased shoot fresh weight, increased biomass by 2.4 times higher than control | [65] |

## 2. Conclusions and Future Perspective

Halophytes offer an ecofriendly alternative to meet the food demand of growing populations by re-vegetating saline lands. They can be utilized as an excellent source of biofuel because they contribute to C balance and inhibit greenhouse gas emission problems. Their root exudates contain some metabolites that recruit beneficial microbes in the rhizosphere. Halophytes synthesize phenolics, flavonoids, and a plethora of bioactive compounds that exhibit antioxidant and antimicrobial properties. The microbes associated with halophytes also produce a diverse group of bioactive secondary metabolites that modulate the growth and yield of crops under stress. Halophyte root pieces habilitating halotolerant bacteria may comprise a good source of biofertilizer or a carrier for biofertilizers. Halophytes are tools for understanding the salt tolerance mechanism of plants and for adapting agriculture to climate change. They can be used as cash crops for biosaline agriculture and also for rehabilitation of arid and semiarid regions. Halophytes also act as good phytoremediators for contaminated lands. The need is to have a deeper insight into the cross communication

between halophytes and microbe's secondary metabolites in rhizosphere to understand and decipher mechanism that inculcate biosaline agriculture.

**Author Contributions:** A.B. conceptualization, Editing, and supervision A. in write up validation. All authors have read and agreed to the published version of the manuscript.

**Funding:** This research received no external funding.

**Institutional Review Board Statement:** Not applicable.

**Informed Consent Statement:** Not applicable.

**Data Availability Statement:** Not applicable.

**Conflicts of Interest:** The authors declare no conflict of interest.

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
