# Peer review of "Climate Change Modulates Halophyte Secondary Metabolites to Reshape Rhizosphere Halobacteria for Biosaline Agriculture"

_applsci, doi:10.3390/app13031299_

Round 1
Reviewer 1 Report
The review presented by Asadullah and Asghari Bano aimed to report newly identified bioactive secondary metabolites that might play role in establishing rhizospheric microbe association, elucidated the negative im-pacts of salt stress, and directing growth and yield of halophytes.
The topic is original and relevant and address a specific gap in the field. The review is up to date and well written and the references are appropriate.
Authors should consider start the review with an introduction of 1 or 2 pages related to the subject studied. The way it was developed, the introduction is properly the literature review.
Authors should consider improve the Figure 2 since the image quality is very low. Also, authors should consider improve conclusion section adding future perspectives and new possibilities for studies.
Author Response
Reviewer 1
- Authors should consider start the review with an introduction of 1 or 2 pages related to the subject studied. The way it was developed, the introduction is properly the literature review.
Response: information required are added in the introduction
- Authors should consider improve the Figure 2 since the image quality is very low.
Response: the figure 2 image quality was improved.
Also, authors should consider improve conclusion section adding future perspectives and new possibilities for studies.
Response: conclusion has been rewritten and future perspective added

Reviewer 2 Report
In this review article, authors have described the physiological characteristics and other attributes of halophytes and had made an elementary effort to connect it with the rhizosphere bacterial community profiles and climate change. While all the three components sound interesting, the article does not provide a cross cutting discussion on correlating these three. The paper needs thorough revision, and also, must be revised for English language.
Abstract need to be elaborated. Few sentences that justify the objective and summarizes the content of review should be incorporated.
The complete article is an general essay on halophytes, their mechanism of salt tolerance, halobacteria and their PGP attributes. If this remains the focus of review, it makes more sense. However, without the experimental evidence, role of climate change on halophyte rhizobacterial population should not be promoted.
The title seems to be very opinionated and bold claim, where climate change has been designated as a major factor that drives halophyte’s secondary metabolite profiles, while in text, such discussion is poorly supported with evidences.
Minor issues:
“It is considered as a potential source for oilseed and phytoremediation”… phytoremediation is not a product.
“On the basis of habitat the halophytes are classified into hydrophalophyes and xerohalophytes.” – either define the terms, or delete.
Figure 1. Its not needed, as it directly repeats what is already given in text.
Author Response
Reviewer 2
in this review article, authors have described the physiological characteristics and other attributes of halophytes and had made an elementary effort to connect it with the rhizosphere bacterial community profiles and climate change. While all the three components sound interesting, the article does not provide a cross cutting discussion on correlating these three. The paper needs thorough revision, and also, must be revised for English language.
Response English language has been improved and entire manuscript was checked and corrected.
Abstract need to be elaborated. Few sentences that justify the objective and summarizes the content of review should be incorporated.
Response: the abstract was elaborated and rewritten.
Response: The complete article is a general essay on halophytes, their mechanism of salt tolerance, halobacteria and their PGP attributes. If this remains the focus of review, it makes more sense. However, without the experimental evidence, role of climate change on halophyte rhizobacterial population should not be promoted.
Response: in this review main focus is given on salinity as outcome of climate change. It has also been mentioned in text that salinity lead to change in root exudates secondary metabolites that recruit rhizosphere microbes.
The title seems to be very opinionated and bold claim, where climate change has been designated as a major factor that drives halophyte’s secondary metabolite profiles, while in text, such discussion is poorly supported with evidences.
Response: Climate change factors (In text salinity, temperature, and elevated CO2 concentration): Some recent article related to above mentioned factors have been discussed which tell us that change in these factors lead to release of different kinds of metabolites in root exudates. That recruit specific microbes.
Minor issues:
“It is considered as a potential source for oilseed and phytoremediation”… phytoremediation is not a product.
Response: this paragraph was corrected.
“On the basis of habitat the halophytes are classified into hydrophalophyes and xerohalophytes.” – either define the terms, or delete.
Response: The terms were defined.
Figure 1. I ts not needed, as it directly repeats what is already given in text
Response: figure 1 was removed.

Round 2
Reviewer 1 Report
The authors answered all the questions raised. Therefore, I suggest the publication of this paper.
Reviewer 2 Report
The paper has been adequately revised, and authors have addressed all the concern. It may be accepted.